# Ambulatory Electrocardiographic Monitoring and Ectopic Beat Detection in Conscious Mice

**DOI:** 10.3390/s20143867

**Published:** 2020-07-10

**Authors:** Felke Steijns, Máté I. Tóth, Anthony Demolder, Lars E. Larsen, Jana Desloovere, Marjolijn Renard, Robrecht Raedt, Patrick Segers, Julie De Backer, Patrick Sips

**Affiliations:** 1Center for Medical Genetics, Department of Biomolecular Medicine, Ghent University, 9000 Ghent, Belgium; Felke.Steijns@UGent.be (F.S.); Marjolijn.Renard@UGent.be (M.R.); 2Institute Biomedical Technology, Ghent University, 9000 Ghent, Belgium; MateIstvan.Toth@UGent.be (M.I.T.); LarsEmil.Larsen@UGent.be (L.E.L.); Patrick.Segers@UGent.be (P.S.); 3Department of Cardiology, Ghent University Hospital, 9000 Ghent, Belgium; Anthony.Demolder@UGent.be (A.D.); Julie.DeBacker@UGent.be (J.D.B.); 44BRAIN, Department of Head and Skin, Ghent University, 9000 Ghent, Belgium; Jana.Desloovere@UGent.be (J.D.); Robrecht.Raedt@UGent.be (R.R.)

**Keywords:** ambulatory electrocardiography, mouse, peak analysis, tethered ECG

## Abstract

Ambulatory electrocardiography (AECG) is a primary diagnostic tool in patients with potential arrhythmic disorders. To study the pathophysiological mechanisms of arrhythmic disorders, mouse models are widely implemented. The use of a technique similar to AECG for mice is thus of great relevance. We have optimized a protocol which allows qualitative, long-term ECG data recording in conscious, freely moving mice. Automated algorithms were developed to efficiently process the large amount of data and calculate the average heart rate (HR), the mean peak-to-peak interval and heart rate variability (HRV) based on peak detection. Ectopic beats are automatically detected based on aberrant peak intervals. As we have incorporated a multiple lead configuration in our ECG set-up, the nature and origin of the suggested ectopic beats can be analyzed in detail. The protocol and analysis tools presented here are promising tools for studies which require detailed, long-term ECG characterization in mouse models with potential arrhythmic disorders.

## 1. Introduction

Cardiovascular disease (CVD) is the leading cause of death worldwide [1,2]. A combination of high mortality and morbidity rates has been attributed to common CVDs including coronary artery disease, stroke, heart failure, and arrhythmias, making early detection and treatment of these CVDs imperative [1,2]. One of the most important screening tools for CVD is the resting electrocardiogram (ECG). The obtained signal on an ECG represents a series of heartbeats each consisting of multiple different deflections of the electrical tracing. These include the P-wave (depolarization of the atria), the QRS-complex (depolarization of the ventricles) and the T-wave (repolarization of the ventricles) [3,4]. A trained physician can utilize the ECG to assess many aspects, including the heart rate and rhythm, conduction abnormalities and myocardial disease [3,4]. Despite its many advantages, a common limitation of an ECG is its duration. A standard ECG registers cardiac signals during a short period of 10 s. Transient conditions such as arrhythmias or conduction problems may therefore remain undetected [3,4].

Ambulatory (or continuous) ECG (AECG) measurement overcomes the temporal limitations of the standard ECG by extending the diagnostic capabilities over a prolonged period of time, typically 24 h or 48 h [5]. This extended monitoring period is associated with an improved diagnostic yield [5]. The European Society of Cardiology guidelines suggest that patients who have arrhythmia-related symptoms should receive a 24-h ambulatory ECG, in order to detect potential detrimental ventricular arrhythmia and conduction disorders and preventing SCD [6,7]. The most common finding on AECG are ectopic heart beats: premature atrial complexes (PACs) and premature ventricular complexes (PVCs) originating from the atrial and ventricular myocardium, respectively [8]. AECG allows for quantification of PACs and PVCs, respectively called “PAC or PVC burden” wherein the number of PACs or PVCs is expressed as a percentage of the total number of recorded heartbeats.

In order to be able to study the pathophysiological mechanisms underlying cardiovascular disease, including cardiac rhythm disorders, in an *in vivo* context, experimental preclinical studies have made extensive use of the mouse as a relevant animal model for human disease [9,10,11,12]. Several techniques have already been described to obtain ECG data in mice [9,13]. These include standard needle electrodes, platform ECG systems [14], tethered ECG systems [13,15] and telemetry [16]. Tethered ECG and telemetry best match the AECG in humans as they allow long-term monitoring in conscious, freely-moving mice (Table 1) [13]. If the experimental design requires knowledge of the spatial resolution (origin) of the ectopic beats, a tethered system is the best option as different lead configurations can be assessed by implanting multiple electrodes.

The application of the tethered ECG approach for use in mice has not been described in great detail thus far, with only limited resources available in the literature [13]. Moreover, the assessment of the origin of potential ectopic beats has not been evaluated before. In this paper we describe the optimization of a protocol for the implementation of tethered ECG measurements in mice. We also further highlight the application of automated algorithms which were developed to automatically assess heart rate (HR), RR-interval, heart rate variability (HRV), and ectopic beats in the collected ECG traces.

## 2. Materials and Methods

### 2.1. Mice

Four Male C57BL/6J mice of 10 weeks of age were used for this study. Housing conditions with a controlled temperature (21 °C–22 °C) and relative humidity (40–60%) at a 12 h/12 h light/dark cycle were used, with food and water available ad libitum. All animal procedures were approved by the local ethical committee of Ghent University (ECD18/15) and were conducted in compliance with directive 2010/63/EU.

### 2.2. Surgery and Instrumentation

Prior to surgery, mice were habituated to an in-house developed harness, which is used for the fixation of the ECG electrode connector, for one week. In order to be able to put on the harness, mice were briefly anesthetized (1.0–1.5% isoflurane, 0.5 L/min 100% O_2_). The harness consists of two parts: a corset made of stretchable cotton and a rigid platform made of Velcro (Figure 1). In the corset, two holes were present for the front paws of the mouse. Two pairs of strings sewn to the lateral sides of the corset were pulled through two pairs of holes on the lateral sides of the rigid platform, assuring that this platform is fixed on the back of the mouse at the level of the mid-scapular region.

After one week of habituation to the harness, mice underwent surgery for electrode lead placement. Mice were anesthetized (induction at 5% isoflurane, 0.5 L/min 100% O_2_ and maintenance at 1.0–1.5% isoflurane, 0.5 L/min 100% O_2_) and placed in a prone position on a heating pad set to 37 °C. The consciousness of the mice was checked by means of tail and foot pinch prior to surgery. The harness was removed and approximately one cm^2^ of hair was shaved at the base level of the four limbs and at the mid-scapular region. The bare skin patches were disinfected with a povidone–iodine solution after which a 0.5 cm transverse incision was made uncovering the underlying muscle tissue. Four insulated wire leads were used (AWG37, stainless steel PTFE coated wire; MEDwire, Sigmund Cohn, Mount Vernon, NY, USA), of which approximately five mm of the insulation was removed at the tip and subsequently curled up in order to increase the contact surface. The wire leads were subcutaneously tunneled starting from the incision at the level of the limbs and exiting the body at the level of the mid-scapular incision. After securing the leads to the underlying muscle tissue at the level of the limbs all incisions were closed with silk suture (6-0, Fine Science Tools, Heidelberg, Germany) and incision sites were treated topically with a neomycin/bacitracin anti-bacterial ointment. The exiting ends of all electrode leads were pulled through a central hole in the rigid platform of the harness. Approximately three mm of insulation was removed at the end of all electrode leads before placing them in a Winslow-type connector strip (female side exposed) attached to the rigid platform of the harness (Figure 1). After all electrode leads were placed in the connector strip, the leads were secured by hot glue. Next, both front paws were pulled through the holes of the corset and the strings were attached to the rigid platform. The mice were disconnected from the anesthesia device and 0.03 mg/kg buprenorphine diluted in 0.5 mL saline solution (0.9%) was injected subcutaneously. Finally, mice were placed in an empty cage with soft bedding under an infrared lamp until they were fully awake. Animals were left for four days to recover from surgery.

### 2.3. ECG Recording

After recovery, mice were slightly sedated with isoflurane before connection to the ECG set-up. The connector strip on the rigid platform of the harness was connected to a 4-channel unity gain preamplifier (based on a TL074 SMT Opamp, Texas Instruments, Dallas, TX, USA) by means of Winslow connector pins (267-7400, RS components, Corby, UK). The preamplifier was attached to a swivel system through a 6-channel commutator (Plastics One, Roanoke, VA, USA) allowing free movement of the animal in the cage. Next, using a custom-made amplifier (based on TL074 Opamp, Texas Instruments), the ECG signals were high-pass filtered at 0.15 Hz and amplified 512 times. The ECG signals were digitized at a sampling rate of 2000 Hz (16-bit resolution, +/− 10 V input range) by means of a NiDAQ card (USB-6259, National Instruments, Austin, TX, USA) with 32 analog input channels. For this study, mice remained connected to the ECG set-up for four consecutive days. Three simultaneously recorded ECG traces were obtained in lead I, II and III configurations (Figure 2). In this study, the ECG data were subdivided into fragments of twenty minutes for the subsequent optimization and validation of the analysis parameters. For analysis of long-term ECG recordings, we developed custom algorithms that can be applied to multiple recorded fragments using a straightforward batch processing mode.

### 2.4. Pre-Processing ECG Trace

Lead II configuration resulted in the strongest signal and was therefore selected for further analysis. We developed custom MATLAB^®^ scripts, which are made available to the community (https://github.com/tothmate9/ecg-analysis), to automate the processing as well as the qualitative and quantitative analysis of the recorded data. Similar to human AECG recordings, pre-processing of the obtained murine ECG data was performed in order to improve the signal quality (reduce baseline wandering and increase signal-to-noise ratio) [17]. However, compared to the pre-processing of human ECG data, an adapted filtering strategy needed to be applied due to the specific characteristics of the obtained murine electrophysiological signal [12] (Figure 3). A first pre-processing step, a zero-phase high-pass filter (HPF), was applied to increase the signal-to-noise ratio [18]. Therefore, a second order Butterworth filter with cut-off frequency of 5 Hz was applied, filtering out respiratory activity and muscle activity [19,20]. After the high-pass filtering, the signal was down-sampled in order to reduce the computational and data storage load for analysis of long data recordings. The optimal down-sampling ratio was determined to retain a maximal amount of information. Next, Fast Fourier Transformation (FFT) was applied in order to increase the signal-to-noise ratio. By means of FFT, the temporal information of the ECG signal is converted into frequency information. A band-pass filter was applied to identify the frequency of the QRS complex and to eliminate the motion artefacts and the baseline wander. For a sample rate of 1000 Hz, the cut-off frequencies were set to 100 and 200 Hz. After band-pass filtering the ECG signal is again converted into standard temporal information by means of inversed FFT [21,22,23]. Next, a zero-phase derivative filter was applied to further increase the signal-to-noise ratio. In contrast to conventional noise reducing filters, the zero-phase filtering does not create a phase shift, hence preserving the temporal resolution of the QRS complex [24,25]. The derivative filter produced a suitable input for subsequent R-peak detection.

### 2.5. ECG Parameter Extraction

First, R-peak detection was performed on the pre-processed signal based on a minimum peak distance and peak height which were determined empirically (default thresholds are 50 ms and 0.05 mV, respectively). The detected R-peaks were displayed on the band-pass filtered ECG-trace. The sensitivity (Equation (1)), precision (Equation (2)) and specificity (Equation (3)) of the R-peak detection of the developed script were compared to the open access platform PhysioZoo which was developed to analyze ECG recordings from humans as well as different mammalian species, including mice [26]. On selected previously analyzed 20-min ECG traces, all R-peaks were manually annotated and considered as reference values needed to calculate the number of true positive, true negative, false positive and false negative peaks detected by the algorithm.
Sensitivity = # true positive/(# true positive + # false negative),(1)
Precision = # true positive/(# true positive + # false positive),(2)
Specificity = # true negative/(# true negative + # false positive),(3)

Next, the different ECG parameters, including average HR, mean RR-interval and HRV, were automatically calculated based on the detected R-peaks. The average HR was computed as the total amount of detected R-peaks on the pre-processed ECG signal divided by the recording time. For the mean RR-interval, the distance in time (ms) between two consecutive R-peaks was determined for all detected R-peaks in the analyzed ECG trace, after which the mean was automatically calculated.

Finally, the RR-intervals detected in the analyzed ECG trace were used for the evaluation of HRV. A histogram was plotted based on the frequency of discrete RR-interval ranges within the analyzed ECG trace. The x-axis represents the observed RR-intervals (within intervals of 1 ms), while the y-axis represents the frequency of the respective RR-interval range. The width of the histogram of all RR-interval ranges at the level of half of the maximum counted prevalence is used as a representative value for HRV, indicated with FWHM (Full-Width at Half-Maximum). This measure provides a read-out regarding the distribution of the normal RR-intervals with larger values indicating a wider heart rate variability. Interference by aberrant RR-intervals due to incorrect R-peak detection and ectopic beats is not included in this method of HRV measurement as the prevalence of their time-intervals is negligible and is most-likely located at the tails of the histogram.

### 2.6. Ectopic Beat Detection

Arrhythmic events were identified semi-automatically based on deviating RR-intervals. We identified the ectopic beats by applying a moving average filter with a 100 RR-interval sliding window, which was empirically determined to provide the best performance. This filter operates on the RR-intervals and marks the detected R-peaks which differ, based on a given threshold, from the moving average as ectopic beats. This threshold is arbitrary and can be manually changed in the script. For this study we used a threshold of 30% as this value is small enough to trigger ectopic beat identification without detection of intervals within the range of normal variability in sinus rhythm [27]. The detected potential ectopic beats were saved as a separate output file for further manual evaluation. In order to assess the type of ectopic beat (for example, PAC versus PVC), manual evaluation of the ECG traces was performed on the unfiltered ECG trace at 2000 Hz. Furthermore, an estimation of the origin of the ectopic beat within the heart was evaluated using all three lead configurations obtained from the different electrode leads.

### 2.7. Statistics

Statistical analysis was performed with the statistical software package SPSS (version 25.0). The comparison of the sensitivity, precision and specificity of the R-peak detection between the custom MATLAB^®^ script and the open access platform PhysioZoo was performed by means of a paired samples test. Correlation analysis between the number of false positive ectopic beat detections and the quality of the ECG trace was performed by implementation of a parametrical Pearson correlation test. A *p*-value < 0.05 was defined as statistically significant (two-sided).

## 3. Results

### 3.1. ECG Recording

During data collection, the harness proved to be stable and allowed for reliable recordings while mice (*n* = 4) were able to move around in their cage without disconnecting from the ECG set-up. Four days after initial connection to the ECG set-up, the data were collected and analyzed offline. Murine ECG traces were obtained using lead I, II and III configurations (Figure 2). On the obtained traces the different deflections, including the P-wave, the QRS-complex, and the T-wave could be distinguished (Figure 2). To evaluate the quality of the ECG trace, time segments on which individual R-peaks could not be manually distinguished were added up and divided by twenty minutes in order to get a percentage. Overall, an average of 0.95% (SE 0.47%) of the traces were of bad quality most likely due to excessive movement of the animal (*n* = 10 ECG traces of twenty minutes).

### 3.2. ECG Parameter Extraction

The sensitivity, precision and specificity of the R-peak detection by our custom automated algorithms were evaluated for ten ECG fragments of twenty minutes randomly selected from four mice after down-sampling the recorded data to 1000 Hz. For this analysis the ECG data obtained from lead II configuration was selected as this trace represented the strongest signal. The obtained values were compared with the output obtained from the PhysioZoo ECG software. No significant difference could be observed in sensitivity, specificity or precision between both methods (Table 2, paired samples test, *p* ≥ 0.144), validating the performance of our novel algorithms. The subsequent calculations of the different ECG-parameters, including average HR, mean RR-interval and HRV, highly depend on this R-peak detection. These values are automatically calculated and presented after analysis of the recorded ECG data by our custom MATLAB^®^ scripts (Figure 4).

Similar measurements of sensitivity, precision and specificity were performed on an ECG recording at different sample rates (2000, 1000, 500, 400 and 250 Hz) in order to evaluate the ideal balance between a maximal reduction in computational load due to data down-sampling while achieving minimal loss of performance for long-term measurements (more than twenty minutes). For this optimization an ECG file with a high-quality signal was used (with only 0.03% of the ECG trace marked as bad quality). We found that the sensitivity of the R-peak detection was dependent on the sample rate of the ECG file (Figure 5), declining significantly with sample rates < 400 Hz. Based on this comparison, for automatic analysis of long-term ECG recordings down-sampling to frequencies below 400 Hz is not recommended.

### 3.3. Ectopic Beat Detection

Analysis of potential ectopic beats was performed on eight twenty-minute ECG traces in lead II configuration. Sample rates of 1000 Hz were used to diminish computational load while preserving high-quality data as ectopic beat detection is based on aberrant RR-intervals. A high sensitivity of R-peak detection is thus warranted as this will lead to less false positive ectopic beat detections. After running the script, potential ectopic beats are visualized as regions of interest (ROI) in a separate output file (pdf format). These pdf files can then be manually checked for the presence and nature of an ectopic beat (Figure 6a). As expected, false positive recordings were also found (Figure 6b). The number of false positive detections significantly correlated with the quality of the ECG trace as determined by the percentage of bad quality tracing (Pearson correlation = 0.731, *p* = 0.039). On the other hand, the number of detected true ectopic beats does not correlate with the quality of the ECG trace (Pearson correlation = 0.172, *p* = 0.685).

In order to analyze the type and estimate the origin of the observed ectopic beats, the ECG recording of the respective beats at 2000 Hz must be used (unfiltered ECG data). Using these high-resolution tracings, the P-waves can also be discerned, which are more difficult to distinguish from background noise when lower sample rates are used. The identification of P-waves can be useful in the evaluation of the type of ectopic beat. Distinction between a blocked P-wave and sinus arrest can only be made based on the presence or absence of a P-wave before a missing QRS-complex, respectively (Figure 7). In the former, a P wave is present without a subsequent QRS-complex, whereas in the latter a P-wave is absent. The probable spatial origin of an observed ectopic beat can be determined depending on the orientation and magnitude of the extra QRS complex in the three obtained lead configurations. Representative examples are presented in Figure 8.

## 4. Discussion

AECG is a powerful diagnostic tool to study cardiac arrhythmia. This is mainly due to its ability to analyze long-term ECG data, which is associated with an improved diagnostic yield [5]. Mice have been widely used as a model system in cardiovascular research mainly because their genome can be easily altered and because there is a strong resemblance to basic human physiology [9]. To investigate cardiac arrhythmia, tethered ECG set-ups allow long-term ECG recordings in conscious, freely moving mice, similar to AECG in humans. In this paper we provided a detailed description of the surgical technique and the instrumentation necessary for a tethered ECG set-up in mice. Furthermore, automated scripts were developed, optimized, and validated for analysis of the obtained ECG data and ectopic beat detection.

The surgical implantation of the electrode leads was similar to a previously published protocol [13], with several important improvements. A custom-made, two-piece harness was made to accommodate the ECG connection to the tether. The customizable, snug fit assured that the harness stays in place during recordings without hampering normal movement or breathing of the mouse. Furthermore, the harness can easily be connected and disconnected to the tether when needed. During recording, no detachment of the electrode leads nor harness was observed in all four included mice for this study. However, as recording periods were limited to four days for this study, the durability of the tethered setup needs to be tested for longer recording periods.

Our tethered ECG set-up was able to generate high-fidelity ECG traces in lead I, II and III configurations. The different murine ECG traces could be visualized and the R-peak could be visually distinguished from the background noise in more than 99% of the ECG traces. ECG segments of bad quality were most likely caused by movement artefacts which are also observed in human AECG recordings [5]. Movement artefacts are considered as an inherent disadvantage of AECG. However, the benefits of AECG recording transcend this disadvantage as certain types of arrhythmia may be transient and can thus only be detected with AECG [5].

As a large amount of data can be obtained with this tethered ECG set-up, automated analysis of the ECG data is necessary for efficient and unbiased data extraction. For this reason, custom MATLAB^®^ scripts were developed. Parameters used for data filtration were optimized, enabling a robust signal-to-noise ratio for efficient automated R-peak detection. Data sample rates were adjusted to find an ideal balance between signal quality and reduced computing load for the analysis of long-term recordings. We were able to confirm that our algorithms have a similar efficiency of R-peak detection compared to PhysioZoo [26], an online available ECG analysis software program specifically tailored to processing ECG recordings from different species including mice. Furthermore, based on this R-peak detection, important ECG parameters including the average HR, the mean RR-interval and HRV were automatically calculated after running the custom MATLAB^®^ scripts on the ECG data. Though these parameters can also be calculated by implementation of the PhysioZoo software, the PhysioZoo software mainly focuses on HRV analysis. For this reason, the pre-filtering of the ECG data in PhysioZoo removes ectopic beats as these interfere with the HRV analysis [26]. This specific pre-filtering step thus hampers subsequent ectopic beat detection which is not the case with our custom scripts. In order to still provide a measure of HRV when using our algorithms, we have included the FWHM measurement of the RR-interval histogram.

Ectopic beat detection relies on sensitive and accurate R-peak detection. Aberrant RR-intervals based on a deviation of the moving average were identified as ROI for ectopic beats. Using this automated annotation of specific ROI’s large amounts of ECG data can be analyzed quickly for the presence of ectopic beats. The quantification of ectopic beats is of interest as several studies have demonstrated an association between high PAC burden and arrhythmias such as atrial fibrillation, atrial flutter and supraventricular tachycardia [28,29]. Likewise, a high PVC burden may be associated with reduced left ventricular ejection fraction, PVC-induced cardiomyopathy and increased risk of sudden cardiac death [30,31].

A unique advantage of the tethered ECG method described in this paper is that multiple lead configurations can be obtained: lead I, II and III. In the absence of abnormal heart orientation, multiple lead configurations allow, to a certain extent, the estimation of the spatial origin of the observed ectopic beat. The knowledge of the origin of an ectopic beat might benefit the search for the underlying disease mechanism in mouse models developing arrhythmia (e.g., myocardial infarction). Furthermore, the assessment of multiple lead configurations can be beneficial to validate the presence of abnormal ECG patterns observed in a single lead. As a future extension, based on these lead configurations the augmented lead configurations (aVR, aVL and aVF) can be calculated. Combined with the lead I, II and III configurations, these augmented leads will give more detailed information about the ECG in the frontal plane of the heart.

## 5. Conclusions

Overall, the described tethered ECG protocol allows long-term recording of high-quality ECG traces in conscious, freely moving mice. Furthermore, automated algorithms are provided for efficient HR, RR-interval and HRV analysis and ectopic beat detection. This technique can be readily applied for the characterization of existing and new mouse models with potential arrhythmic disorders. Furthermore, this set-up can be used to test the effects of anti-arrhythmic drugs as well as to assess cardiovascular safety of novel drugs which could potentially be translated to the clinic.

## 6. Limitations

Using the current script, stretches of bad quality data within the input ECG data are not automatically excluded for subsequent HR analysis. Consequently, the calculated HR will slightly differ from the true HR. However, due to prior extensive pre-processing of the ECG signal, the percentage of bad quality stretches is limited and will only have a negligible effect on the calculation of the HR. Furthermore, we were unable to prove the ability of the developed script to detect more complex patterns of activation than PVC and PAC, as these events are rarely found in mice [12] and did not occur in the wild-type mice used in our study.

## Figures and Tables

**Figure 1 sensors-20-03867-f001:**
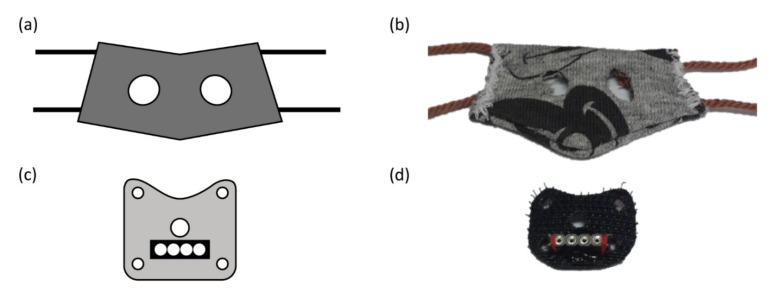
Schematic representation and picture of the in-house developed harness. (**a**,**b**) Corset piece with cut-outs for the front paws and two pairs of strings attached to the lateral side. (**c**,**d**) Rigid platform with central hole for the electrode leads and one hole at each corner for connection to the strings of the corset. Black rectangle represents connector strip (female side exposed) with four insertion places for each electrode lead.

**Figure 2 sensors-20-03867-f002:**
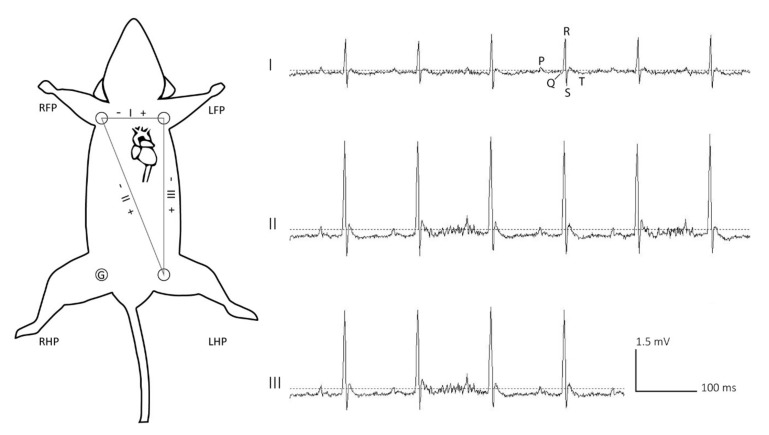
Mouse in supine position with schematic representation of the obtained lead configuration (lead I, lead II and lead III) with representative ECG-traces. P-wave, QRS complex, and T-wave are annotated on the ECG-trace in lead I configuration. Vertical and horizontal scale bars represent 1.5 mV and 100 ms, respectively. G; ground, RFP; right front paw, LFP; left front paw, RHP; right hind paw, LHP; left hind paw.

**Figure 3 sensors-20-03867-f003:**
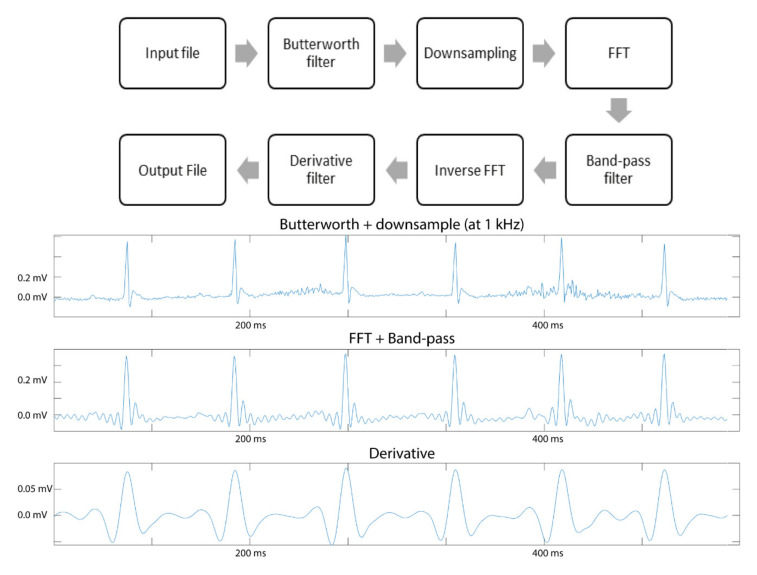
Schematic representation of the different pre-processing steps applied to the input file and resulting effect on the ECG trace. FFT; Fast Fourier Transformation.

**Figure 4 sensors-20-03867-f004:**
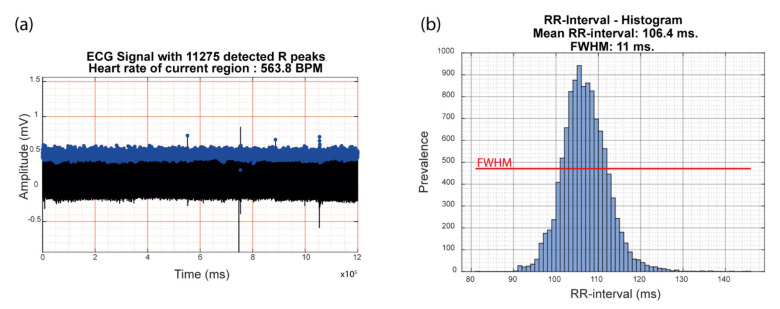
Graphical user interface presentation of the automatically calculated ECG parameters after running the custom MATLAB^®^ scripts on a representative twenty-minute murine ECG trace. (**a**) Detected R-peaks are visualized with blue dots on the (compressed) murine ECG trace. Total amount of detected R-peaks and calculated heart rate is presented in the title of the graph. (**b**) Histogram representing the distribution of the observed RR-intervals. X-values are subdivided in intervals of 1 ms. Automated calculation of the mean RR-interval and Full-Width at Half-Maximum (representative for heart rate variability) is presented in the title of the graph. The level of the Full-Width at Half-Maximum value is visualized by means of a red line within the graph. ECG; electrocardiogram, BPM; beats per minute, FWHM; Full-Width at Half-Maximum.

**Figure 5 sensors-20-03867-f005:**
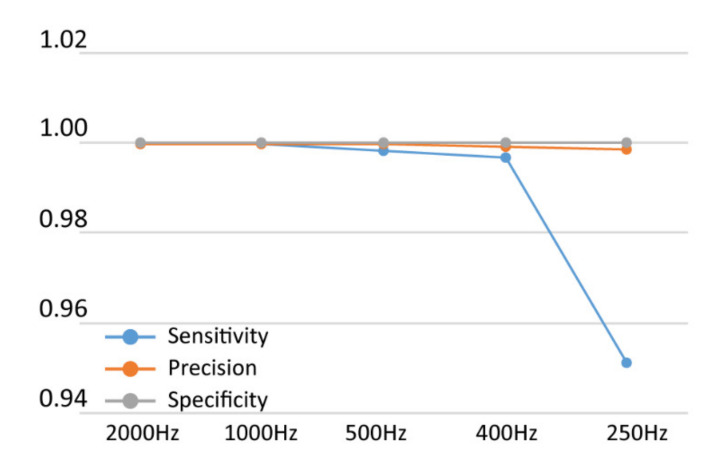
Sensitivity, precision and specificity of R-peak detection of the same ECG file at different sample rates analyzed with our custom algorithms.

**Figure 6 sensors-20-03867-f006:**
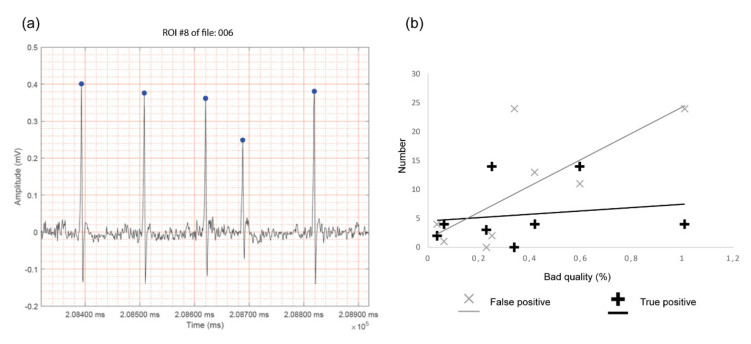
Ectopic beat detection. (**a**) Representative output obtained for ectopic beat detection showing the aberrant RR-interval. Detected R-peaks are indicated with blue dots on the band-pass filtered ECG-trace. (**b**) Correlation analysis between the percentage of bad quality of the ECG trace and false positive and true positive ectopic beat detection (*n* = 8 ECG traces of twenty minutes). The number of false positive and true positive ectopic beats are depicted with a grey x and a black +, respectively. ROI; Region of interest.

**Figure 7 sensors-20-03867-f007:**
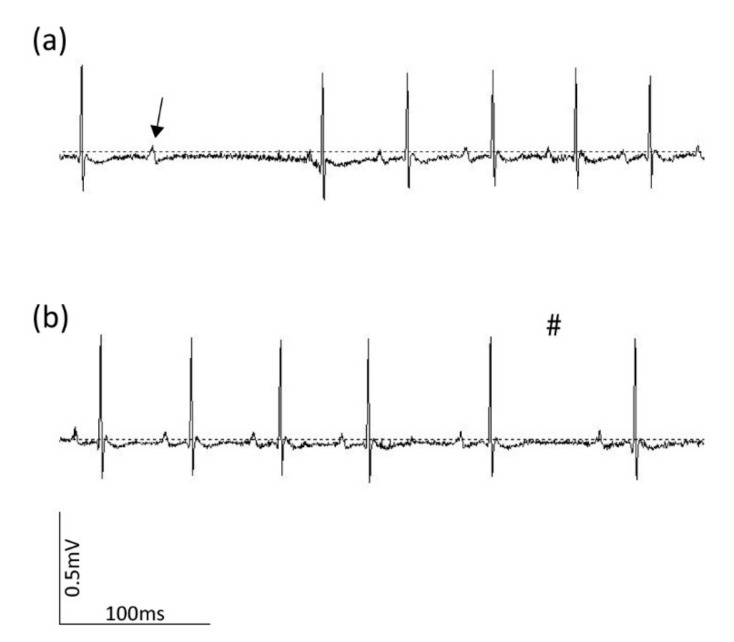
P-wave analysis on representative mouse ECG files at a sample rate of 2000 Hz. (**a**) Blocked P-wave based on the presence of a P-wave (indicated with arrow) without subsequent QRS-complex. (**b**) Sinus arrest based on the absence of an extra P-wave in between a long RR-interval. Vertical and horizontal scale bars represent 0.5 mV and 100 ms, respectively. Hashtag indicates absence of p-wave indicating sinus arrest.

**Figure 8 sensors-20-03867-f008:**
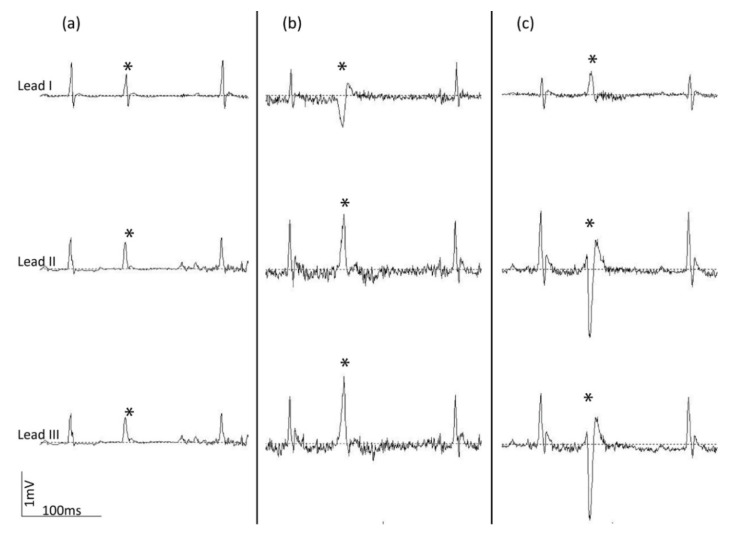
Representative ectopic beats recorded in mice using lead I, II and III configurations. (**a**) An ectopic beat with either an atrial or junctional origin, based on a narrow QRS complex; (**b**,**c**) ectopic beats with origin in the ventricles (based on a broad QRS complex); for (**b**) an inferior axis and for (**c**) a superior axis can be inferred based on the ECG pattern. Vertical and horizontal scale bars represent 1 mV and 100 ms, respectively. Ectopic beats are indicated with asterisks.

**Table 1 sensors-20-03867-t001:** Characteristics of human ambulatory electrocardiography (AECG) and ECG techniques available in mice.

	Human AECG	Mouse ECG			
		Needle Electrodes	Platform ECG System	Tethered ECG System	Telemetry
Conscious study subject	+	−	+	+	+
Long-term analysis	+	−	−	+	+
Unrestrained	+	−	−	+/−	+
Spatial resolution ectopic beat	+	+	+	+	−

ECG: electrocardiography; AECG: ambulatory electrocardiography.

**Table 2 sensors-20-03867-t002:** Sensitivity, precision and specificity of R-peak detection compared between our automated custom algorithms and PhysioZoo (*n* = 10 ECG traces of twenty minutes).

	File	TP	TN	FP	FN	Sensitivity	Precision	Specificity
CUSTOM ALGORITHMS	001	11780	1188216	2	2	0.999830	0.999830	0.999998
	002	12413	1187556	16	15	0.998793	0.998713	0.999987
	003	11274	1188723	1	2	0.999823	0.999911	0.999999
	004	11209	1188758	18	15	0.998664	0.998397	0.999985
	005	10524	1189468	4	4	0.999620	0.999620	0.999997
	006	11580	1188405	8	7	0.999396	0.999310	0.999993
	007	10794	1189197	2	7	0.999352	0.999815	0.999998
	008	11085	1188871	25	19	0.998289	0.997750	0.999979
	009	13366	1186524	56	54	0.995976	0.995828	0.999953
	010	13411	1186489	45	55	0.995916	0.996656	0.999962
			Mean values			0.998566	0.998583	0.999985
PhysioZoo	001	11773	1188209	11	9	0.999236	0.999067	0.999991
	002	12417	1188484	3	11	0.999115	0.999758	0.999997
	003	11270	1187595	6	6	0.999468	0.999468	0.999995
	004	11222	1188730	3	2	0.999822	0.999733	0.999997
	005	10521	1189491	9	7	0.999335	0.999145	0.999992
	006	11574	1188412	7	13	0.998878	0.999396	0.999994
	007	10785	1189199	11	16	0.998519	0.998981	0.999991
	008	11072	1188864	29	32	0.997118	0.997388	0.999976
	009	13348	1186497	73	72	0.994635	0.994561	0.999938
	010	13410	1186419	53	56	0.995841	0.996063	0.999955
			Mean values			0.998197	0.998356	0.999983
			*p*-values			0.144	0.408	0.360

TP: true positive; TN: true negative; FP: false positive; FN; false negative.

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
