# Peer review of "Ambulatory Electrocardiographic Monitoring and Ectopic Beat Detection in Conscious Mice"

_sensors, 2020, doi:10.3390/s20143867_

Round 1

Reviewer 1 Report

The work presented complies with the best scientific standards, in addition to three incremental levels of complexity. First of all, authors devote a relevant activity to in-vivo biological subjects, along with the need for a comprehensive set of customization efforts as a prerequisite for further research. In this first step it was develop of harness elements, surgery of the subjects under analysis, and the training and adaptation process of the subjects.

Secondly, it has been necessary to adapt and develop personalized solutions from a sensors device point of view for ECG recordings. And finally, they applied the necessary signal processing toward an ultimate purpose of detecting arrhythmic events.

Taking into account the wide effort presented here, and although under the area of ​​knowledge of this reviewer, this very last point related to signal processing could be improved by adding comparative analysis with human recordings methods, the extensive activity presented here it is justified by itself for publication without any further effort.

This reviewer therefore wishes to congratulate the research team and encourage them to continue in this direction, incorporating, if not in this, in future works, comparative analysis of the results between the animal and the human case.

Reviewer 2 Report

This a paper about technological improvements in ECG recordings in mice. As the authors state in the introduction, this is an important topic for study arrhythmia mechanisms in experimental studies.

Main limitation is the unproven ability of the system for detection of more complex patterns of activation than PVC/PAC.

Authors overestimate the ability of limb leads to localize the origin of PVC in the ventricles.  Morphological criteria based on leads I, II and III, as displayed in figure 8, can just make an approximation. I disagree with the red dots positioned at the schematic representations of the heart.

Reviewer 3 Report

The paper deals with a very interesting topic of an ambulatory ECG monitoring and ectopic beat detection algorithm in mice. A unique harness was developed for long-term ECG monitoring without the restraint of mouse movement. The arrangement of harness, commutator and amplifier provides high-quality ECG signals. The method of ectopic beat detection is proposed and clearly described. The result of the detection is the identification of ROIs with ectopic beats. 

Although the methods are clearly described, some parts should be described in more details:

The FFT filtering was used in signal preprocessing. The values of cut-off frequencies of band-pass filter are not specified.

Table 2 summarizes sensitivity, precision and specificity of R-peak detection. How the TP, TN, FP and FN values were determined?

Round 2

Reviewer 2 Report

I appreciate author`s feedback to my comments. I agree with issue number 1, but figure 8 still need for edition. I still disagree with authors regarding the potential origing of PVCs. I will suggest authors to focus on the ability to discern between inferior vs superior axis (will be better to display a parasternal axis). May be authors should request assistance from cardiologist.
